# Wuqinxi Qigong as an Alternative Exercise for Improving Risk Factors Associated with Metabolic Syndrome: A Meta-Analysis of Randomized Controlled Trials

**DOI:** 10.3390/ijerph16081396

**Published:** 2019-04-18

**Authors:** Liye Zou, Yangjie Zhang, Jeffer Eidi Sasaki, Albert S. Yeung, Lin Yang, Paul D. Loprinzi, Jian Sun, Shijie Liu, Jane Jie Yu, Shengyan Sun, Yuqiang Mai

**Affiliations:** 1Faculty of Athletic Training, Guangzhou Sport University, Guangzhou 510500, China; liyezou123@gmail.com (L.Z.); sjian@vip.sohu.com (J.S.); 2Lifestyle (Mind-Body Movement) Research Center, College of Sports Science, Shenzhen University, Shenzhen 518060, China; 3Health and Exercise Science Laboratory, Institute of Sports Science, Seoul National University, Seoul 08826, Korea; elite_zhangyj@163.com; 4Department of Sport Sciences, Institute of Health Sciences, Federal University of Triangulo Mineiro, Uberaba, MG 38025-440, Brazil; jeffersasaki@gmail.com; 5Depression Clinical and Research Program, Harvard Medical School, Boston, MA 02114, USA; ayeung@mgh.harvard.edu; 6Cancer Epidemiology and Prevention Research, Alberta Health Services, Calgary, AB T2S 3G3, Canada; lin.yang@ahs.ca; 7Departments of Oncology and Community Health Sciences, Cunning School of Medicine, University of Calgary, Calgary, AB T2N 4N1, Canada; 8Department of Health, Exercise Science and Recreation Management School of Applied Sciences, The University of Mississippi, Oxford, MS 36877, USA; pdloprin@olemiss.edu; 9Department of Physical Education, Wuhan University of Technology, Wuhan 430070, China; liushijie0411@whut.edu.cn; 10Sports and Exercise Psychology Laboratory, Department of Sports, Science and Physical Education, The Chinese University of Hong Kong, Shatin, New Territories, Hong Kong, China; yujie@cuhk.edu.hk; 11Institute of Physical Education, Huzhou University, Huzhou 313000, China; 12College of Chinese Martial Arts, Guangzhou Sport University, Guangzhou 510500, China

**Keywords:** exercise, Qigong, mind–body exercise, metabolic syndrome, risk factor

## Abstract

*Background:* The improvement of living standards has led to increases in the prevalence of hypokinetic diseases. In particular, multifactorial complex diseases, such as metabolic syndrome, are becoming more prevalent. Currently, developing effective methods to combat or prevent metabolic syndrome is of critical public health importance. Thus, we conducted a systematic review to evaluate the existing literature regarding the effects of Wuqinxi exercise on reducing risk factors related to metabolic syndrome. *Methods:* Both English- and Chinese-language databases were searched for randomized controlled trials investigating the effects of Wuqinxi on these outcomes. Meanwhile, we extracted usable data for computing pooled effect size estimates, along with the random-effects model. *Results:* The synthesized results showed positive effects of Wuqinxi exercise on systolic blood pressure (SBP, *SMD* = 0.62, 95% CI 0.38 to 0.85, *p* < 0.001, *I*^2^ = 24.06%), diastolic blood pressure (DBP, *SMD* = 0.62, 95% CI 0.22 to 1.00, *p* < 0.001, *I*^2^ = 61.28%), total plasma cholesterol (TC, *SMD* = 0.88, 95% CI 0.41 to 1.36, *p* < 0.001, *I*^2^ = 78.71%), triglyceride (TG, *SMD* = 0.87, 95% CI 0.49 to 1.24, *p* < 0.001, *I*^2^ = 67.22%), low-density lipoprotein cholesterol (LDL-C, *SMD* = 1.24, 95% CI 0.76 to 1.72, *p* < 0.001, *I*^2^ = 78.27%), and high-density lipoprotein cholesterol (HDL, *SMD* = 0.95, 95% CI 0.43 to 1.46, *p* < 0.001, *I*^2^ = 82.27%). In addition, regression results showed that longer-duration Wuqinxi intervention significantly improved DBP (*β* = 0.00016, *Q* = 5.72, df = 1, *p* = 0.02), TC (*β* = −0.00010, *Q* = 9.03, df = 1, *p* = 0.01), TG (*β* = 0.00012, *Q* = 6.23, df = 1, *p* = 0.01), and LDL (*β* = 0.00011, *Q* = 5.52, df = 1, *p* = 0.02). *Conclusions:* Wuqinxi may be an effective intervention to alleviate the cardiovascular disease risk factors of metabolic syndrome.

## 1. Introduction

Undeniably, the improvement of living standards has led to increases in the prevalence of hypokinetic diseases. In particular, multifactorial complex diseases, such as metabolic syndrome [1], are becoming more prevalent. Metabolic syndrome, as a cluster of risk factors (e.g., abdominal obesity, high triglycerides, blood glucose and blood pressure, and low high-density lipoprotein (HDL) cholesterol), is strongly related to the risk for cardiovascular disease, atherosclerosis, and diabetes [2,3], as well as higher risk for cardiovascular mortality and morbidity [4]. According to population-based statistics with older adults in a Rotterdam study, prevalence estimates of metabolic syndrome in adults range from 19% to 42%, based on the definition of metabolic syndrome [4]. Currently, developing effective methods to combat or prevent metabolic syndrome is of critical public health importance.

Epidemiological studies have shown that a healthy lifestyle is a contributor to reduce the prevalence of metabolic syndrome in adults [5]. Among the diversity of lifestyle habits, regular physical activity is closely associated with the prevention of the metabolic syndrome. A cross-sectional study demonstrated that adults engaging in 23 MET·week^−1^ of physical activity (MET, metabolic equivalent of task) have a low prevalence of metabolic syndrome [6]. Conversely, adults who engage in prolonged sedentary behavior are at risk for various metabolic syndrome constituents, including low HDL cholesterol and large abdominal circumference [7]. A comprehensive review by Ford and Li demonstrated that physical activity may help reduce the risk of metabolic syndrome among adults [8]. Therefore, engaging in physical activity can be effective in the prevention and treatment of metabolic syndrome.

In addition to traditional exercises (e.g., aerobic walking), there is a need and interest in identifying alternative forms of exercise to help prevent and treat metabolic syndrome [9]. Wuqinxi is one traditional Chinese exercise that is starting to receive attention in the research literature (e.g., Tai Chi, Buduanjin, and other Qigong forms) [10,11]. The Wuqinxi exercise routine was originally choreographed by an ancient Chinese physician in the Donghan Dynasty [12]; it is figuratively known as the “five animals” exercise, including movements imitating tigers, deer, bears, apes, and birds [12]. Wuqinxi modalities such as Tai Chi [13,14,15], Baduanjin [16,17,18], and Yoga [19,20,21] also involve symmetrically slow movements, integrated with breathing techniques, physical and mental relaxation, body awareness, and meditation. Since the establishment of the Health Qigong Association in 2004, Wuqinxi and other traditional Chinese health-promoting Qigong exercises have become widely practiced worldwide [22,23,24]. Accumulating evidence suggest that practicing Wuqinxi may actively improve health outcomes in individuals with chronic obstructive pulmonary disease [25], osteoporosis [26], and lumbosacral multifidus [27]. Additionally, studies have begun to investigate the effects of Wuqinxi on risk factors (e.g., triglycerides, blood glucose and blood pressure, and HDL cholesterol) for metabolic syndrome, suggesting that Wuqinxi exercise may help to improve these relevant biomarkers [28,29,30,31,32].

Although the number of Wuqinxi studies on metabolic syndrome have increased, a systematic review is absent so far. Therefore, we conducted a systematic review on this topic, with two aims: (i) to evaluate the effectiveness of Wuqinxi exercise on cardiovascular outcomes related to metabolic syndrome; (ii) to determine whether the long-term practice of Wuqinxi exercise is more effective against metabolic syndrome than non-Wuqinxi control groups. To our knowledge, our study is the first systematic review to assess the potential efficacy of Wuqinxi exercise on cardiovascular biomarkers related to metabolic syndrome. This synthesized review will summarize the current scientific evidence on this topic, which will provide utility for both researchers and health-related professionals.

## 2. Methods

### 2.1. Data Sources

Four English-language databases, namely PubMed, Web of Science, Scopus, and the Cochrane library, were used for the literature search. Given that Wuqinxi Qigong exercise was originated in China, we also searched Chinese-language electronic databases, including China National Knowledge Infrastructure (CNKI), Wanfang, and the Chinese Biomedical Database. The end date of our search was December 18, 2018. The following three groups of keywords, searched individually and in combination with each other, were employed in the search: (i) Wuqinxi, five-animal exercise, five-animal Qigong, or five-animal boxing; (ii) metabolic syndrome, blood lipid, total plasma cholesterol, triglyceride, high-density lipoprotein cholesterol, low-density lipoprotein cholesterol, blood pressure systolic, and diastolic blood pressure. The literature search was performed by one of the coauthors (Y.Z.) and confirmed by another individual (L.Z.), in which no additional articles were identified.

### 2.2. Inclusion Criteria and Study Selection

Studies were included in this systematic review if they: (i) were randomized controlled trials in peer-reviewed journals; (ii) recruited human subjects; (iii) included Wuqinxi Qigong exercise as the main intervention in the experimental group, as compared to an active or non-active control condition; (iv) reported at least one parameter of blood pressure and/or blood lipid (e.g., cholesterol), with obtainable data including mean and standard deviation of each group at baseline and post-intervention, along with the number of participants in each group.

Preliminary screening was conducted by one reviewer (Y.Z.) by examining the title and abstract of the retrieved documents, with the objective of removing (i) duplicates and/or (ii) apparently irrelevant documents. Based on the inclusion criteria, the potentially relevant articles were further assessed by two independent reviewers (Y.Z. and L.Z.), so that we were able to determine the number of eligible studies included in this systematic review. Further, reference lists of reviews and finally selected articles were manually cross-checked. When disagreements between the two reviewers occurred regarding study selection, a third reviewer (S.L.) evaluated the manuscript to reach a final consensus. Study selection process was shown in Figure 1.

### 2.3. Methodological Quality of Included Studies

To independently perform methodological assessments of the included studies, two review authors (Y.Z. and L.Z.) used the Physical Therapy Evidence Database (PEDro) scale. This widely accepted assessment tool for methodological quality consists of 11 items: eligibility criteria, randomization, concealed allocation, baseline equivalence, blinding of stakeholders (participants, instructor/therapist, and assessor), attrition rate of ≤15%, intention to treat analysis, between-group statistical comparison, and point measures/measures of variability. Blinding of participants and instructors are unrealistic during exercise intervention [33]. Thus, these two items were not considered in this systematic review, which resulted in a final number of 9 items, with each item worth one point. A higher score is reflective of a greater methodological quality.

### 2.4. Data Extraction of Included Studies

Detailed information of each selected study was independently extracted by two reviewers (Y.Z. and L.Z.) and converted into two standardized forms: one for descriptive data and another one for quantitative data. Descriptive data included the study reference (first author and publication year), study location and language of publication, participant characteristics (health status, sample size/attribution rate, mean age/age range), intervention protocol, and outcome measured. In addition, we also extracted quantitative data (randomly allocated number of participants and mean/standard deviation of each group at baseline and post-intervention) for calculating effect size estimates of the intervention.

### 2.5. Data Synthesis

We used the Comprehensive Meta-Analysis Software (Bio. Stat. Inc., Englewood, NJ, USA) for synthesizing the extracted data. The template (means, pre- and post-treatment SD, sample size in each group) was selected, along with pre/post-treatment correlation of 0.5 [33]. Standardized mean difference (*SMD*) was selected as the present effect size (ES) estimate in all pooled analyses. It is worth emphasizing that data of control condition and Wuqinxi intervention were placed on the left and right side, respectively. Positive value of *SMD* and *β* indicates that Wuqinxi has favorable effects on these selected outcomes. A random-effect model with 95% confidence interval (CI) was employed to calculate the ESs for each outcome measure. The magnitude of ES was classified as: (i) 0.2 = small; (ii) 0.5 = medium; (iii) 0.8 = large [34]. I-squared (25% = small, 50% = medium, and 75% = large) test was selected to determine heterogeneity across studies. Publication bias for all outcomes was evaluated using the Egger’s test. In addition, meta-regression was used to investigate the continuous variable (total training time) with a mixed-effects model.

## 3. Results

### 3.1. Study Selection

A total of 149 studies indexed in the aforementioned electronic databases were identified. Additionally, we retrieved four studies from reference lists of review papers. Thus, 153 studies were initially retrieved, with 33 duplicates removed. The remaining 120 studies were further screened according to the titles and abstracts, which resulted in 33 records left. Full-text articles were assessed against the predetermined inclusion criteria, and nine articles were finally included.

### 3.2. Features of Selected Randomized Controlled Trials

All nine RCTs (randomized controlled trials) were conducted in different provinces of China and published between 2009 and 2016 (Table 1).

In total, this review included 628 participants with different health status. The number of participants (with age range between 35 and 75 years or mean age of 52.6 to 62.11 years) in each study ranged from 30 to 110, with an attrition rate of 10.3% [35] and 11.4% [38] reported in two studies. Of the nine studies, eight reported Wuqinxi as the exercise intervention alone, while a combination of Wuqinxi plus drug therapy was reported in one study [36]. Furthermore, intervention duration lasted 8 to 48 weeks; each session lasted 30–60 min, with 3–7 sessions per week. No adverse events occurred during the Wuqinxi exercise intervention. None of the studies conducted a follow-up assessment.

### 3.3. Methodological Quality Assessment

The quality of eligible studies was presented in Table 2. Three studies [35,39,43] had a study quality score of 6, while six studies [36,37,38,40,41,42] had a score of 7. None of the studies used concealed allocation and blinding of assessors. In addition, baseline equivalence was absent in three studies [35,39,43].

### 3.4. The Synthesis Effectiveness of Wuqinxi

Studies investigated the effects of Wuqinxi exercise on six outcomes (SBP (systolic blood pressure), DBP (diastolic blood pressure), TC (total plasma cholesterol), TG (triglycerides), LDL (low-density lipoprotein), and HDL (high-density lipoprotein)) (Table 3). The synthesized results showed positive effects of Wuqinxi exercise on SBP (*SMD* = 0.62, 95% CI 0.38 to 0.85, *p* < 0.001, *I*^2^ = 24.06%; Figure 2), DBP (*SMD* = 0.62, 95% CI 0.22 to 1.00, *p* < 0.001, *I*^2^ = 61.28%; Figure 3), TC (*SMD* = 0.88, 95% CI 0.41 to 1.36, *p* < 0.001, *I*^2^ = 78.71%; Figure 4), TG (*SMD* = 0.87, 95% CI 0.49 to 1.24, *p* < 0.001, *I*^2^ = 67.22%; Figure 5), LDL (*SMD* = 1.24, 95% CI 0.76 to 1.72, *p* < 0.001, *I*^2^ = 78.27%; Figure 6), and HDL (*SMD* = 0.95, 95% CI 0.43 to 1.46, *p* < 0.001, *I*^2^ = 82.27%; Figure 7).

### 3.5. Moderator Analysis

In order to examine the effect of Wuqinxi exercise time, meta-regression was performed to determine if the total training time influenced the different indices in Table 4. Regression results showed that longer-duration Wuqinxi intervention significantly improved DBP (*β* = 0.00016, *Q* = 5.72, df = 1, *p* = 0.02), TC (*β* = −0.00010, *Q* = 9.03, df = 1, *p* = 0.01), TG (*β* = 0.00012, *Q* = 6.23, df = 1, *p* = 0.01), and LDL (*β* = 0.00011, *Q* = 5.52, df = 1, *p* = 0.02). There were no significant relationships between total training time and Wuqinxi exercise on SBP (*β* = −0.00001, *Q* = 0.41, df = 1, *p* = 0.52) and HDL (*β* = 0.00005, *Q* = 1.12, df = 1, *p* = 0.29).

## 4. Discussion

This systematic review critically evaluated and statistically synthesized the evidence on the effects of Wuqinxi exercise on individual biomarkers associated with metabolic syndrome. Based on the available evidence, our review suggests that this traditional mind–body exercise is effective in the treatment of six of the metabolic syndrome constituents (SBP, DBP, TC, TG, LDL, and HDL). The standardized mean difference ranged between 0.62 and 1.24. Thus, the magnitudes of effects were at least moderate, with large effects observed for most variables (4 out of 6). This has significant public health importance as metabolic syndrome is on the rise in many parts of the world and it is associated with the risk of developing cardiovascular disease and type 2 diabetes [44]. While the exact mechanisms of how Wuqinxi affects metabolic syndrome constituents are unknown, the findings from this study provide support for the use of Wuqinxi as an important adjunct for treatment of metabolic syndrome risk factors. Although speculative, and in need of future research, it is plausible that Wuqinxi may improve cardiovascular risk factors from the physical exercise component of Wuqinxi as well as from the psychological (which may manifest physiologically) benefits associated with this type of exercise. Wuqinxi exercises are accessible to people of all ages and physical strength, easy to learn, and have little known side effects. Thus, it should be scientifically promoted for more people to be involved, to maintain health and wellbeing. The results of each metabolic syndrome risk factor are described below.

### 4.1. Blood Pressure

The results suggest that Wuqinxi practice is effective in reducing both systolic and diastolic blood pressure. The overall effects of Wuqinxi on systolic and diastolic blood pressure were of medium magnitude. In regards to systolic blood pressure, five of the six studies presented results favoring Wuqinxi, when compared to the control condition (usual care) [35,40,41,42,43]. These studies included samples ranging from 30 to 110 participants, mostly middle-aged adults [35,40,41,42,43]. The training protocols involved weekly frequencies ranging between 3 and 6 days per week, with training sessions lasting for 60 min [35,40,41,42,43]. The total duration of the interventions ranged between 24 and 48 weeks [35,40,41,42,43]. The only study where no significant group differences in systolic blood pressure were observed utilized a 12-week intervention [36]. Thus, the results suggest that duration of the training program may be essential for inducing favorable effects of Wuqinxi on systolic blood pressure.

As for diastolic blood pressure, four of the five studies demonstrated favorable results for Wuqinxi when compared to control conditions (usual care, usual care + drug therapy, and Liuzijue) [36,40,41,42]. In these studies, training sessions lasted for 60 min and training frequency ranged between 3 and 6 days per week [36,40,41,42]. Regarding the training program, the minimum duration for achieving favorable results of Wuqinxi on diastolic blood pressure was 12 weeks [40]. The only study not showing significant results for Wuqinxi utilized a training frequency of 5 days per week during a 24-week period [43]. This study did not indicate the duration of the sessions [43].

Overall, the results suggest that practicing Wuqinxi 3–6 days per week for 60 min/day, for a period of at least 12–24 weeks, can lead to significant reductions in both systolic and diastolic blood pressure.

### 4.2. Total Plasma Cholesterol

Five of the six studies included in our meta-analysis demonstrated positive effects of Wuqinxi on total plasma cholesterol [35,36,37,38,39]. The magnitude of effect was large, as indicated by the standardized mean difference. Sample sizes for the statistically significant studies were between 30 and 104 participants, mostly middle-aged adults [35,36,37,38,39]. These studies utilized a training frequency of 5–7 days per week with similar session durations of 60 min, except for the study of Li et al. [39], in which training sessions lasted for 30 min. The training program duration ranged between 8 and 48 weeks [35,36,37,38,39]. The only study that did not demonstrate significant results in favor of Wuqinxi included a sample size of 40 participants, with an age range of 60–75 years [41]. Training frequency, training session, and intervention duration for this study were 6 days/week, 60 min, and 24 weeks, respectively [41]. Taken together, the results of these studies suggest that practicing 60 min of Wuqinxi, 5–7 days per week, for at least 8 weeks, can lead to lower levels of total plasma cholesterol.

### 4.3. Triglycerides

All six evaluated studies demonstrated significant results for Wuqinxi in lowering triglycerides levels [36,37,38,39,40,41]. The magnitude of this effect was large. These studies included samples ranging between 30 and 100 participants, mostly middle-aged adults [36,37,38,39,40,41]. The control condition for four of the studies employed a usual care condition [37,38,40,41]. On the other hand, one study utilized usual care + drug therapy [36], and another study used walking training [39] as the control condition. Training frequency in the studies ranged between 5 and 7 days per week, with training sessions lasting for 60 min, except for the study of Li et al. [39], in which the training sessions were only 30 min long. The evaluated studies included interventions lasting between 8 and 24 weeks [36,37,38,39,40,41]. Thus, these results suggest that as little as 30 min of Wuqinxi performed 5 days per week, for a minimum of 8 weeks, can lead to significant reductions in triglycerides.

### 4.4. Low-Density and High-Density Lipoprotein Cholesterol

Among the dependent variables examined in this meta-analysis, LDL cholesterol was the one for which Wuqinxi training presented the greatest effect, with an overall standardized mean difference of 1.24, which is classified as a large effect. For HDL cholesterol, the effects of Wuqinxi was less pronounced, but still considered large (*SMD* = 0.95). For LDL cholesterol, all six studies demonstrated significant results in favor of Wuqinxi when compared to the control conditions (usual care, usual care + drug therapy, or walking training) [36,37,38,39,40,41]. The only exception for HDL cholesterol was the study by Li et al. [39], which did not demonstrate significant results for Wuqinxi compared to the control condition. The characteristics of the training protocol were the same as cited above for the triglycerides variable, as the studies included were the same [36,37,38,39,40,41]. This meta-analysis suggests that 30–60 min of Wuqinxi performed 5–7 days per week, for a minimum of 8–12 weeks, is able to promote reductions in LDL cholesterol and increases in HDL cholesterol.

### 4.5. Moderator Effects of Wuqinxi on the Dependent Variables

The moderator analysis demonstrated that the effects of Wuqinxi on DBP, TC, TG, and LDL cholesterol were partially moderated by total training time. However, the moderating effects of total training time on the dependent variables was low, as depicted by the β coefficients of the regressions. A possible explanation for the weak effects of total training time on the dependent variables is the low number of studies meta-analyzed and thus the small variability in total training time. It is possible that the moderating effects would be higher had more studies been identified and included in the analysis.

### 4.6. Remarks

Interestingly, the studies selected in this meta-analysis were all performed in China, possibly due to the relative lack of familiarity with Wuqinxi in Western countries. Other strengths of this study include the use of a recognized meta-analytic method to evaluate the magnitude of Wuqinxi intervention effects (pooled effect size), the analyses of potential factors that could influence outcomes of Wuqinxi practices (moderator analyses), and the extent of heterogeneity across studies using *I*^2^ statistics. Nonetheless, the following methodological limitations should be acknowledged as they may influence the interpretation of our research findings. One of the most important drawbacks among the studies on this topic is that concealment of the intervention was not possible, as subjects knew whether or not they received Wuqinxi training, even though centralized randomization was used. This might lead to subjectivity and expectation bias by the participants. Second, Wuqinxi was not offered as a monotherapy in one of the selected studies, but as an adjunctive treatment. It may be difficult to definitely conclude whether the outcomes were due to Wuqinxi alone, to a synergetic intervention effect, or to the conventional treatment received by the patients. Third, it is unclear why treatment effects did not differ by duration of Wuqinxi practice in the moderator analysis. It is possible that in most studies, participants had attained the minimal duration needed to obtain the benefits. The frequency and session length may be less important given that Wuqinxi is relatively easy to learn and that the majority of the benefits might have come from daily self-practice and not from the training sessions. Fourth, not all of the studies evaluated all metabolic syndrome risk factors (e.g., abdominal obesity).

## 5. Conclusions

This systematic review, based on the existing literature, suggests that Wuqinxi may be an effective intervention to alleviate the cardiovascular disease risk factors of metabolic syndrome. More RCTs with rigorous research design are warranted to establish the therapeutic effects of Wuqinxi for metabolic syndrome risk factors and its potential to be used in healthy lifestyle intervention programs for prevention or treatment of metabolic syndrome and other related chronic diseases.

## Figures and Tables

**Figure 1 ijerph-16-01396-f001:**
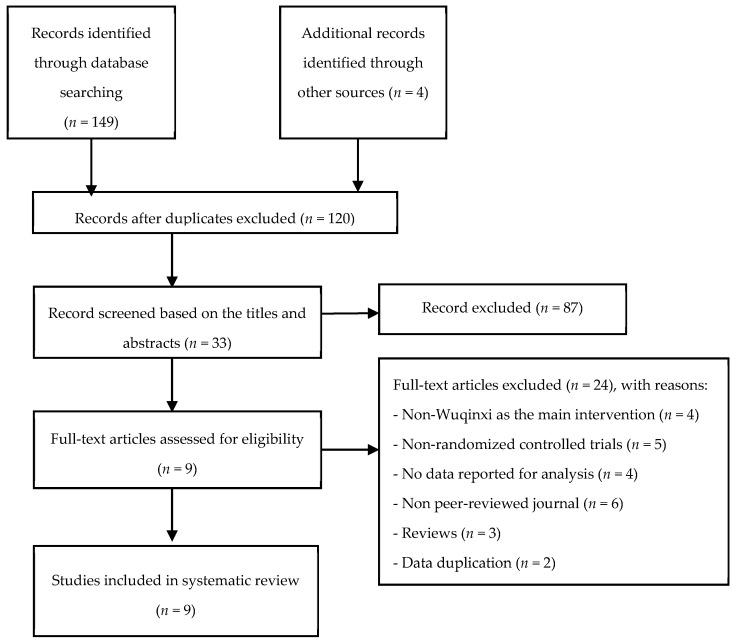
Flowchart displaying study selection.

**Figure 2 ijerph-16-01396-f002:**
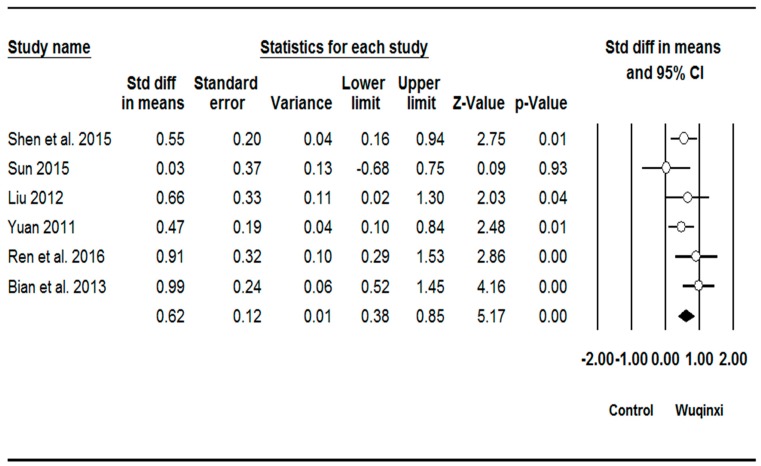
Effect of Wuqinxi on systolic blood pressure. Effect size (95% CI) of each study is denoted by circle (black line). The mid-point of the circle represents the point effect estimate, meaning effect estimate for each study. The area of the circle represents the weight given to the study. The diamond below the studies represents the overall effect.

**Figure 3 ijerph-16-01396-f003:**
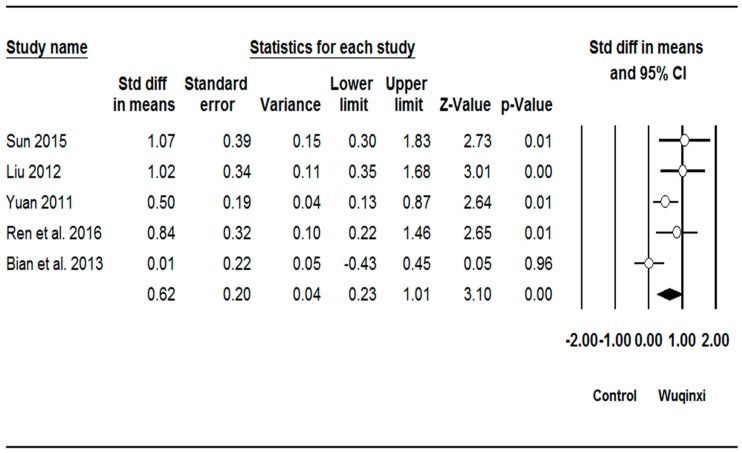
Effect of Wuqinxi on diastolic blood pressure. Effect size (95% CI) of each study is denoted by circle (black line). The mid-point of the circle represents the point effect estimate, meaning effect estimate for each study. The area of the circle represents the weight given to the study. The diamond below the studies represents the overall effect.

**Figure 4 ijerph-16-01396-f004:**
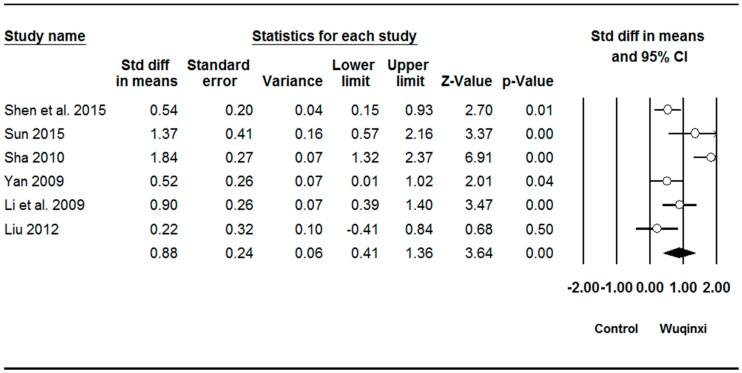
Effect of Wuqinxi on total plasma cholesterol. Effect size (95% CI) of each study is denoted by circle (black line). The mid-point of the circle represents the point effect estimate, meaning effect estimate for each study. The area of the circle represents the weight given to the study. The diamond below the studies represents the overall effect.

**Figure 5 ijerph-16-01396-f005:**
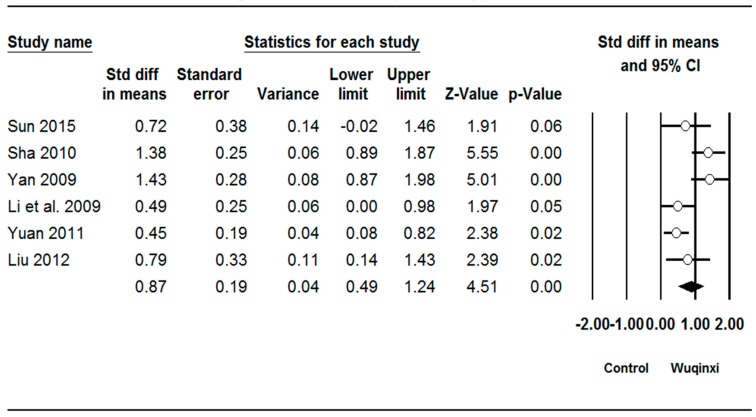
Effect of Wuqinxi on triglycerides. Effect size (95% CI) of each study is denoted by circle (black line). The mid-point of the circle represents the point effect estimate, meaning effect estimate for each study. The area of the circle represents the weight given to the study. The diamond below the studies represents the overall effect.

**Figure 6 ijerph-16-01396-f006:**
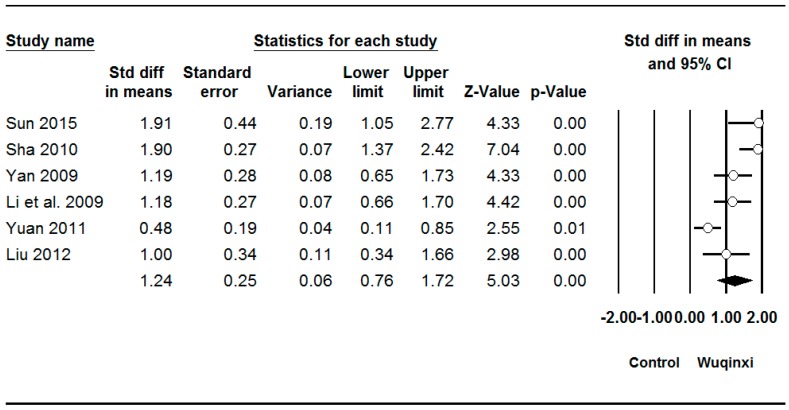
Effect of Wuqinxi on low-density lipoprotein cholesterol. Effect size (95% CI) of each study is denoted by circle (black line). The mid-point of the circle represents the point effect estimate, meaning effect estimate for each study. The area of the circle represents the weight given to the study. The diamond below the studies represents the overall effect.

**Figure 7 ijerph-16-01396-f007:**
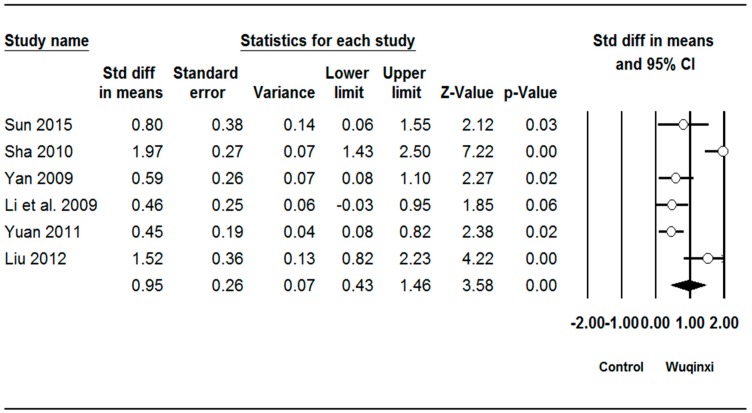
Effect of Wuqinxi on high-density lipoprotein cholesterol. Effect size (95% CI) of each study is denoted by circle (black line). The mid-point of the circle represents the point effect estimate, meaning effect estimate for each study. The area of the circle represents the weight given to the study. The diamond below the studies represents the overall effect.

**Table 1 ijerph-16-01396-t001:** Summary of included studies.

Reference	Location (Language)	Participant Characteristics		Intervention Program	Wuqinxi Training			Outcome Measured	Adverse Event; Follow-Up
		Sample Size (Attrition Rate)	Mean Age or Age Range		Frequency (weekly)	Time (min)	Duration (week)		
Shen et al. (2015) [35]	Jiangsu, China (Chinese)	104 community residents (10.3%)	35–59	EG: Wuqinxi CG: usual care	5	60	48	Blood lipid parameter (TC); Blood pressure index (SBP)	No; No
Sun (2015) [36]	Shandong, China (Chinese)	30 people with metabolic syndrome (0%)	40–50	EG: Wuqinxi + DT CG: usual care + DT	5	60	24	Blood lipid parameters (TC, TG, HDL-C, LDL-C); Blood pressure index (SBP, DBP)	No; No
Sha (2010) [37]	Changan, China (Chinese)	80 community residents (0%)	WQ: 57.78 ± 5.50 CG: 57.68 ± 5.46	EG: Wuqinxi CG: usual care	5	60	20	Blood lipid parameters (TC, TG, HDL-C, LDL-C)	No; No
Yan (2009) [38]	Dalian, China (Chinese)	70 people with hypertriglyceridemia (11.4%)	WQ: 52.6 ± 8.70 CG: 52.6± 8.70	EG: Wuqinxi CG: usual care	7	60	24	Blood lipid parameters (TC, TG, HDL-C, LDL-C)	No; No
Li et al. (2009) [39]	Guangzhou, China (Chinese)	66 people with hyperlipoidemia (0%)	WQ: 58.67 ± 20.43 CG: 56.47 ± 24.15	EG: Wuqinxi CG: walking training	7	30	8	Blood lipid parameters (TC, TG, HDL-C, LDL-C)	No; No
Yuan (2011) [40]	Shandong, China (Chinese)	110 community residents (0%)	WQ: 61.40 ± 1.57 CG: 62.11 ± 1.53	EG: Wuqinxi CG: usual care	5	60	12	Blood lipid parameters (TC, TG, HDL-C, LDL-C) Blood pressure index (SBP, DBP)	No; No
Liu (2012) [41]	Shanxi, China (Chinese)	40 people with metabolic syndrome (0%)	60–75	EG: Wuqinxi CG: usual care	6	60	24	Blood lipid parameters (TC, TG, HDL-C, LDL-C)	No; No
Ren et al. (2016) [42]	Xian, China (Chinese)	44 community residents (0%)	WQ: 61.07 ± 2.63 CG: 57.31 ± 1.65	EG: Wuqinxi CG: usual care	3	60	24	Blood pressure index (SBP, DBP)	No; No
Dian et al. (2013) [43]	Guangzhou, China (Chinese)	84 community residents (0%)	WQ: 61.26 ± 4.27 CG: 61.52 ± 4.05	EG: Wuqinxi CG: usual care	5	-	24	Blood pressure index (SBP, DBP); cardiac function (SV, HR)	No; No

Note: WQ = Wuqinxi; EG = experimental group; CG = control group; DT = drug therapy; TC = total plasma cholesterol; TG = triglyceride; HDL-C = high-density lipoprotein cholesterol; LDL-C = low-density lipoprotein cholesterol; SBP = systolic blood pressure; DBP = diastolic blood pressure; SV = stroke volume; HR = heart rate.

**Table 2 ijerph-16-01396-t002:** Study quality assessment for eligible randomized controlled trials.

Reference	Item 1	Item 2	Item 3	Item 4	Item 5	Item 6	Item 7	Item 8	Item 9	Score
Shen et al. (2015) [35]	1	1	0	0	0	1	1	1	1	6
Sun (2015) [36]	1	1	0	1	0	1	1	1	1	7
Sha (2010) [37]	1	1	0	1	0	1	1	1	1	7
Yan (2009) [38]	1	1	0	1	0	1	1	1	1	7
Li et al. (2009) [39]	1	1	0	0	0	1	1	1	1	6
Yuan (2011) [40]	1	1	0	1	0	1	1	1	1	7
Liu (2012) [41]	1	1	0	1	0	1	1	1	1	7
Ren et al. (2016) [42]	1	1	0	1	0	1	1	1	1	7
Bian et al. (2013) [43]	1	1	0	0	0	1	1	1	1	6

Note: Item 1 = eligibility criteria; Item 2 = randomization; Item 3 = concealed allocation; Item 4 = baseline equivalence; Item 5 = blinding of assessor; Item 6 = attrition rate of ≤15%; Item 7 = intention to treat analysis; Item 8 = between-group statistical comparison; Item 9 = point measures/measures of variability; 1 = explicitly described and present in details; 0 = absent, inadequately described, or unclear.

**Table 3 ijerph-16-01396-t003:** Synthesized results for the effects of Wuqinxi versus control.

Outcomes	Number of Comparisons	Sample Size	Meta-Analysis			Heterogeneity			Publication Bias
			*SMD*	95% CI	*p*-Value	*I*^2^ %	*Q*-Value	df(*Q*)	Egger’s Test (*p*)
SBP	6	412	0.62	0.38 to 0.85	0.00	24.06%	6.58	5	0.97
DBP	5	308	0.62	0.22to 1.00	0.00	61.28%	10.33	4	0.16
TC	6	390	0.88	0.41 to 1.36	0.00	78.71%	23.48	5	0.61
TG	6	396	0.87	0.49 to 1.24	0.00	67.22%	15.25	5	0.48
LDL	6	396	1.24	0.761 to 1.72	0.00	78.27%	23.00	5	0.12
HDL	6	396	0.95	0.43 to 1.46	0.00	82.27%	28.20	5	0.30

Note: TC = total plasma cholesterol; TG = triglycerides; HDL = high-density lipoprotein; LDL = low-density lipoprotein; SBP = systolic blood pressure; DBP = diastolic blood pressure.

**Table 4 ijerph-16-01396-t004:** Moderator analysis for the effects of Wuqinxi versus control intervention (continuous predictor).

Outcomes	Continuous Predictors	Number of Comparisons	*β*	95% CI	*Q*-Value	df (*Q*)	*p*-Value
SBP	Total training time	6	−0.00001	−0.00006 to 0.00003	0.41	1	0.52
DBP	Total training time	5	0.00016	0.00003 to 0.00028	5.724	1	0.02
TC	Total training time	6	−0.00010	−0.00016 to −0.00003	9.03	1	0.01
TG	Total training time	6	0.00012	0.00003 to 0.00021	6.23	1	0.01
LDL	Total training time	6	0.00011	0.00002 to 0.00020	5.52	1	0.02
HDL	Total training time	6	0.00005	−0.00004 to 0.00014	1.12	1	0.29

Note: TC = total plasma cholesterol; TG = triglycerides; HDL = high-density lipoprotein; LDL = low-density lipoprotein; SBP = systolic blood pressure; DBP = diastolic blood pressure.

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
