# Peer review of "Wuqinxi Qigong as an Alternative Exercise for Improving Risk Factors Associated with Metabolic Syndrome: A Meta-Analysis of Randomized Controlled Trials"

_ijerph, 2019, doi:10.3390/ijerph16081396_

Round 1

Reviewer 1 Report

The investigation titled “Wuqinxi Qigong as an Alternative Exercise for Improving Risk Factors Associated with Metabolic Syndrome: A Meta-Analysis of Randomized Controlled Trials” performed a comprehensive review of the literature in the English and Chinese languages, isolating only randomized controlled trials with the aim of summarizing the current scientific evidence. The manuscript is generally well written with the results well presented. Still, there are some components of this study that should be addressed prior to publication.

Comments

1.     The premise of the study surround metabolic syndrome. However, metabolic syndrome is not accurately defined at the onset (Alberti 2009), and the discussion mentions SBP, DBP, TC, TG, LDL, and HDL as six constituents of metabolic syndrome. The authors may need to retitle to manuscript to more accurately reflect that cardiometabolic risk factors are being investigators and not metabolic syndrome. Particularly given the limited discussion on metabolic syndrome.

2.     Abdominal obesity and total fasting glucose are not investigated as part of the surveyed RCTs, again limiting this investigations reliance to metabolic syndrome.

3.     There is no indication of how many total subjects were included in pooled moderator analysis.

4.     Although mentioned as a limitation, Sun et al 2015 employed Wuqinxi was an adjunctive therapeutic intervention and may not be appropriate to include in the analysis.

5.     Although it’s acknowledged that there are no known mechanisms of how Wuqinxi improves cardiometabolic risk factors, the authors should hypothesize about the underlying mechanisms in the discussion section.

6.     There are some minor formatting errors in tables that need to be addressed.

Author Response

Reviewer 1

The investigation titled “Wuqinxi Qigong as an Alternative Exercise for Improving Risk Factors Associated with Metabolic Syndrome: A Meta-Analysis of Randomized Controlled Trials” performed a comprehensive review of the literature in the English and Chinese languages, isolating only randomized controlled trials with the aim of summarizing the current scientific evidence. The manuscript is generally well written with the results well presented. Still, there are some components of this study that should be addressed prior to publication.

Comments

1.     The premise of the study surround metabolic syndrome. However, metabolic syndrome is not accurately defined at the onset (Alberti 2009), and the discussion mentions SBP, DBP, TC, TG, LDL, and HDL as six constituents of metabolic syndrome. The authors may need to retitle to manuscript to more accurately reflect that cardiometabolic risk factors are being investigators and not metabolic syndrome. Particularly given the limited discussion on metabolic syndrome.

Response: Thank you. You bring up a good point. After careful review of our title, if perceived as acceptable, we would like to retain our current title. Reason is that our title refers to “…risk factors associated with metabolic syndrome.” Since it references “risk factors”, we feel that it accurately captures our paper.

2.    Abdominal obesity and total fasting glucose are not investigated as part of the surveyed RCTs, again limiting this investigations reliance to metabolic syndrome.

Response: Thank you. We have made note of this in the Limitations section.

3.     There is no indication of how many total subjects were included in pooled moderator analysis.

Response: Thanks for your comments. We have added the total subjects about the moderator analysis in corresponding Table.

 4.    Although mentioned as a limitation, Sun et al 2015 employed Wuqinxi was an adjunctive therapeutic intervention and may not be appropriate to include in the analysis.

Response: Thanks for your comments. Because there were 30 people with Metabolic syndrome, in reality/from an ethical perspective, it is reasonable that Wuqinxi is integrated with other main treatment. Thus, we included it.

5.     Although it’s acknowledged that there are no known mechanisms of how Wuqinxi improves cardiometabolic risk factors, the authors should hypothesize about the underlying mechanisms in the discussion section.

Response: Thank you. We have revised the discussion section to hypothesize potential mechanisms.

6.  There are some minor formatting errors in tables that need to be addressed.

Response: Thank you. We have reviewed each table carefully.

Reviewer 2 Report

This is a well-written manuscript with good logical flow of thought from the solid literature background discussions to the final conclusions. Enjoyed reading the rationale that formed the basis for the two research questions. 

There are a few minor grammatical edits suggested to make the writing even stronger. In addition, it is suggested that additional information might be added in the context to explain some of the abbreviations along the way. 

Your statistical methods were sound and appropriate based on the research question. 

Good use of tables to present summary information to support conclusions. 

Thanks for your research contribution to the field. 

Author Response

Reviewer 2

This is a well-written manuscript with good logical flow of thought from the solid literature background discussions to the final conclusions. Enjoyed reading the rationale that formed the basis for the two research questions. 

Response: Thanks for your comments!

There are a few minor grammatical edits suggested to make the writing even stronger. In addition, it is suggested that additional information might be added in the context to explain some of the abbreviations along the way. 

Response: Thank you. The manuscript has been revised accordingly.

Your statistical methods were sound and appropriate based on the research question. 

Good use of tables to present summary information to support conclusions. Thanks for your research contribution to the field. 

Response: Thank you very much for your comments!